# Association between COVID-19 Vaccines and Menstrual Disorders: Retrospective Cohort Study of Women Aged 12–55 Years Old in Catalonia, Spain

**DOI:** 10.3390/ijerph21081090

**Published:** 2024-08-18

**Authors:** Laura Esteban-Cledera, Carlo Alberto Bissacco, Meritxell Pallejá-Millán, Marcela Villalobos, Felipe Villalobos

**Affiliations:** 1Fundació Institut Universitari per a la Recerca a l’Atenció Primària de Salut Jordi Gol i Gurina (IDIAPJGol), 08007 Barcelona, Spain; laura.estebancl@gmail.com (L.E.-C.); cabissacco@idiapjgol.info (C.A.B.); mpalleja@idiapjgol.info (M.P.-M.); 2Departament de Ciències Experimentals i de la Salut, Universitat Pompeu Fabra, 08003 Barcelona, Spain; 3Estudis de Ciències de la Salut, Universitat Oberta de Catalunya, 08012 Barcelona, Spain; mvillalobosm@uoc.edu

**Keywords:** COVID-19 vaccine, menstrual disorders, women health

## Abstract

During the rapid development of COVID-19 vaccines, concerns emerged about potential adverse effects on menstrual health. This study examines the association between COVID-19 vaccination—considering the number of doses and vaccine type—and menstrual disorders, specifically heavy menstrual bleeding (HMB) and amenorrhea (AM). Utilizing electronic health records from the Sistema d’Informació per al Desenvolupament de la Investigació en Atenció Primària (SIDIAP) database in Catalonia, Spain, the retrospective cohort included 1,172,621 vaccinated women aged 12–55 with no prior menstrual disorders observed from 27 December 2020 to 30 June 2023. The incidence rate of HMB and AM increased with the second and third doses of the vaccine. Notably, the AstraZeneca^®^ and Janssen^®^ vaccines were associated with higher odds of HMB (OR: 1.765, CI: 1.527–2.033; OR: 2.155, CI: 1.873–2.476, respectively) and AM (OR: 1.623, CI: 1.416–1.854; OR: 1.989, CI: 1.740–2.269, respectively) from the first to the second dose compared to Pfizer/BioNTech^®^. Conversely, the Moderna^®^ vaccine appeared to offer a protective effect against HMB (OR: 0.852, CI: 0.771–0.939) and AM (OR: 0.861, CI: 0.790–0.937) between the second and third doses. These results were adjusted for potential confounders, such as age, previous COVID-19 infection, and other relevant covariates.

## 1. Introduction

In the context of the global COVID-19 pandemic, declared by the World Health Organization on 11 March 2020, the rapid mobilization to develop and deliver vaccines has been an unprecedented scientific achievement [1]. However, the speed of development and scale of vaccination have raised questions about adverse effects, including women of reproductive age [2]. A possible link between COVID-19 vaccination and menstrual disorders was noted due to several individual case safety reports that emerged during the mass vaccination campaign. In 2022, the European Medicines Agency (EMA) informed that the mRNA vaccines may be associated with the occurrence of menstrual disorders, specifically heavy menstrual bleeding (HMB) and amenorrhea (AM) [3,4]. This situation generated not only great scientific interest but also concern in society, as menstruation is a crucial indicator of female health [5,6].

Various cross-sectional and retrospective cohort studies, including women aged 18–55 years old, reported that menorrhagia, metrorrhagia, and polymenorrhea were the most observed problems after COVID-19 vaccination [7]. In addition, cases of menstrual cycle changes in women emerged, so it was thought that the COVID-19 vaccine could affect menstrual health [8]. The association between the type of COVID-19 vaccine, the doses administered, and menstrual disorders is controversial. Some studies showed that menstrual disorders are independently associated with the type of COVID-19 vaccine used [9,10]. However, other studies reported an association with the Pfizer/BioNTech^®^, Moderna^®^, and AstraZeneca^®^ COVID-19 vaccines and after the second dose [11,12,13,14]. Other authors did not find any association [11,15,16,17]. Menstrual characteristics are dynamic and can vary throughout a woman’s life due to factors such as age, hormonal changes, and transitions like puberty or menopause [8]. Individuals who menstruate regularly may also experience ovulation disorders that are sporadic or induced by external factors such as age, weight, physical activity, diet, caffeine consumption, smoking, exposure to organic solvents, occupation, workplace, medical conditions, medication, and lifestyle factors [18,19], thereby complicating the assessment of potential vaccine-related effects.

This study aims to contribute to the knowledge of the association between COVID-19 vaccination and menstrual disorders, using information from public health system records from Catalonia, Spain. Our objective is to retrospectively examine whether there is an association between COVID-19 vaccination and two specific menstrual disorders, HMB and AM, in women aged 12 to 55 years who have no prior history of menstrual disorders. In addition, given the dynamic and varied nature of menstrual characteristics influenced by a range of factors, our study included covariables such as age, socioeconomic status, chronic health conditions, lifestyle habits, past COVID-19 infections, and hormonal medication use to adjust for potential confounders. By addressing the limitations of previous studies regarding methodology, sample size, and geographical diversity [20], this research seeks to provide a more comprehensive analysis of this public health issue.

## 2. Materials and Methods

### 2.1. Study Design, Setting, and Data Sources

This is a retrospective cohort study using collected electronic health records (EHRs) from the Sistema d’Informació per al Desenvolupament de la Investigació en Atenció Primària (SIDIAP; www.sidiap.org, accessed on 01 May 2024) database in Catalonia, Spain, from 27 December 2020 (first dose of COVID-19 vaccine identified in SIDIAP) to 30 June 2023 (date of the last data availability).

SIDIAP is a primary care database set up by the Primary Care Research Institute of Research (IDIAP Jordi Gol) and the Catalan Institute of Health. The database collects information from 278 primary healthcare centers and includes almost 8 million patients. SIDIAP data comprise the clinical and referral events registered by primary healthcare professionals and administrative staff in EHRs, comprehensive demographic information, community pharmacy invoicing data, specialist referrals, primary care laboratory test results, and vaccines. SIDIAP can also be linked to other data sources, such as the hospital discharge database. Health professionals gather this information using ICD-10-CM codes, ATC codes, and structured forms designed for the collection of variables relevant to primary care clinical management [21]. SIDIAP is listed in the Catalogue of Real World Data sources and studies by EMA ENCePP resources databases (https://catalogues.ema.europa.eu/node/1019/administrative-details, accessed on 01 August 2024).

This study was designed and conducted following the guidelines set forth by the STROBE statement (Strengthening the Reporting of Observational Studies in Epidemiology) to ensure clarity, transparency, and rigor in the collection, analysis, and presentation of the data obtained [22].

### 2.2. Population

We included in the study all women registered in the SIDIAP database who met the following inclusion criteria: (1) aged between 12 and 55 years old at the start of the study period; (2) have received at least one dose of the following COVID-19 vaccines: Pfizer/BioNTech^®^, Moderna^®^, AstraZeneca^®^, and Janssen^®^; and (3) have at least one year of follow-up data available in the database prior to 27 December 2020. Women were excluded from the study if they had a diagnosis of any menstrual disorder prior to COVID-19 vaccination and/or were pregnant and/or breastfeeding.

### 2.3. Variables

#### 2.3.1. Exposure

The exposure of interest was the administration of any doses of the following COVID-19 vaccines: Pfizer/BioNTech^®^, Moderna^®^, AstraZeneca^®^, and Janssen^®^. Accordingly, the brand of the vaccine and the date of vaccination were extracted.

#### 2.3.2. Outcome

The main outcomes were a recorded diagnosis of HMB and AM, defined by using data source-specific code lists (Appendix A) occurring after the vaccination with any dose of COVID-19 vaccines. Individuals experiencing both events contributed to both groups.

#### 2.3.3. Covariates

Information on various covariates was extracted at the start of the study period, including socio-demographic characteristics, such as age (years), and categorized by groups, including 12–18, 19–25, 26–35, 36–45, and 46–55 years old; country of birth (Spain, out of Spain); and socio-economic status measured by the MEDEA deprivation index (Q1–Q5) [23]. Concomitant chronic degenerative diseases were also considered, including diagnosis of type 1 and 2 diabetes mellitus (yes or no), hypertension (yes or no), obesity (yes or no), dyslipidemia (yes or no), auto-immune diseases (yes or no), serious heart condition (yes or no), chronic kidney diseases (yes or no), HIV (yes or no), and polycystic ovary syndrome (yes or no). Each condition was defined based on a combination of diagnosis codes (Appendix A). Lifestyle factors were considered, including alcohol consumption (high risk or low risk), smoking status (smoker, non-smoker, or ex-smoker), sleep duration (daily hours), and body mass index (BMI) (kg/m^2^). For COVID-19 history, the individuals’ COVID-19 records were captured by considering a positive diagnosis code (Appendix A), a positive PCR-RT test, or a positive antigen test before COVID-19 vaccination. Hormonal medication (sex hormones and modulators of the genital system) used (yes or no) before COVID-19 vaccination was extracted using ATC codes (Appendix A).

### 2.4. Statistical Analysis

The descriptive analysis included frequencies (absolute and relative) for categorical variables and means, medians, standard deviation (SD), interquartile range (Q1–Q3), and minimum and maximum values (Min–Max) for continuous variables to examine the data’s dispersion and validity. Bivariate analyses involved specific statistical tests, such as the χ^2^ or Fisher’s F-test, for categorical variables and Student’s *t*-test for continuous variables. Statistical analyses were applied based on the normality of the data.

The incidence rates (IRs) of HMB and AM by dose and type of COVID-19 vaccine were calculated based on the first occurrence of an event between doses or up to the end of the study period for women who did not receive an additional dose and required absence of that event in the year prior (population at risk). The IRs were expressed by the number of new cases per 100,000 individuals.

Logistic regression analyses were conducted to assess the risk of HMB and AM as outcomes following the administration of COVID-19 vaccines as exposures (Pfizer/BioNTech^®^ vaccine as reference). The analyses were adjusted for potential confounding factors, such as age group (12–18 age group as reference), previous diagnosis of COVID-19 disease, type 1 and 2 diabetes mellitus, hypertension, auto-immune disease, obesity, polycystic ovary syndrome, chronic respiratory disease, serious heart condition, and hormonal medication use (yes). The interval of time between administered doses was calculated and considered in the analysis as a potential confounder.

Statistical significance was set at *p*-value < 0.05. All the analyses were performed using the statistical software R version 4.4.0.

## 3. Results

This study analyzed a sample of 1,172,621 vaccinated women aged between 12 and 55 years old without any menstrual disorder prior to COVID-19 vaccination. Within this cohort, 25,014 women had a diagnosis of HMB, and 37,923 women had AM after COVID-19 vaccination. Additionally, 2560 women were diagnosed with both conditions. Women diagnosed with both HMB and AM were included in the analyses for both conditions (Figure 1).

### 3.1. Heavy Menstrual Bleeding

#### 3.1.1. Socio-Demographic Characteristics

The results demonstrated that a higher percentage of women experiencing HMB (26.5%) were born outside of Spain compared to the non-HMB (20.8%) (*p* < 0.001). The age in the HMB group was significantly higher (36.50 years ± 11.25) compared to the non-HMB group (35.61 years ± 12.50) (*p* < 0.001). The differences in age distribution were statistically significant between groups, with the 36 to 45 years age group being the most prevalent (36.3%) (Table 1).

The quintiles with lower socioeconomic status, specifically quintiles 3, 4, and 5, showed a higher prevalence of HMB (22.0%, 22.2%, and 21.6% respectively) compared to their representation in the total population of vaccinated women. In contrast, quintile 1, which represents the more affluent areas, had only 14.4% of the cases of HMB, which is significantly lower than its proportion in the total population (21.0%) (*p* < 0.001) (Table 1).

#### 3.1.2. Chronic Degenerative Diseases

A higher prevalence of various medical conditions was observed in women who experienced HMB post-vaccination. Notably, type 1 and type 2 diabetes were more common in the HMB group, affecting 1.8% of these women compared to only 1.2% in the non-HMB group (*p* < 0.001). Hypertension was also more prevalent in the HMB group, occurring in 5.7% of affected women versus 3.7% in the non-HMB group (*p* < 0.001). Similarly, chronic auto-immune and kidney diseases showed a higher prevalence in the HMB group, with statistically significant differences. Furthermore, polycystic ovary syndrome was more common in women with HMB (1.4% vs. 0.9%, *p* < 0.001) (Table 1)

Additionally, diseases related to metabolic disorders, such as dyslipidemia and obesity, were more frequently present among women with HMB, with prevalences of 3.4% and 4.9% respectively, compared to 2.8% and 3.6% in women without the condition (*p* < 0.001 for both). Chronic respiratory diseases and serious heart diseases were also more common in the HMB group (9.2% vs. 6.5%, *p* < 0.001), although their overall prevalence was low but statistically significant (Table 1).

#### 3.1.3. Hormonal Medication Use

The use of hormonal medication was higher in the HMB group compared to the non-HMB group (28.1% vs. 11.2%, *p* < 0.001) (Table 1).

#### 3.1.4. Lifestyle

No significant differences were observed (Table 1).

#### 3.1.5. History of COVID-19

A higher percentage of women in the HMB group (54.0%) reported having had COVID-19 compared to the non-HMB group (49.7%) (*p* < 0.001) before COVID-19 vaccination (Table 1).

#### 3.1.6. Vaccination Status and Incidence Rate of Heavy Menstrual Bleeding

Minor differences were observed in the number of vaccine doses received between the groups, though most received between two and three doses (Table 1).

For dose 1, out of 1,172,621 vaccinated women, there were 4152 new cases of HMB, resulting in an IR of 330.0 cases per 100,000 inhabitants of the population at risk. Pfizer/BioNTech^®^ vaccines accounted for 2083 cases (IR = 254.3), followed by Moderna^®^, with 883 cases (IR = 426.5), Janssen^®^, with 914 cases (IR = 1428.0), and AstraZeneca^®^, with 272 cases (IR = 330.0) (Table 2).

For dose 2, among 1,063,341 vaccinated women, there were 13,875 new cases of HMB, having an incidence rate of 1304.8 cases per 100,000 inhabitants. The distribution of cases was highest with Pfizer/BioNTech^®^, which had 9987 cases (IR = 1329.1), then Moderna^®^, with 3261 cases (IR = 1351.6), and AstraZeneca^®^, with 625 cases (0.89%). Janssen^®^ reported less than five cases (Table 2).

For dose 3, a total of 534,621 women received this dose, from which 6954 new cases of HMB were reported, giving an incidence rate of 1300.73 cases per 100,000 inhabitants. The majority were from Moderna^®^, with 5828 cases (IR = 1371.24), and Pfizer/BioNTech^®^, with 1125 cases (IR = 1027.15). AstraZeneca^®^ had less than five cases (Table 2).

For dose 4, this dose was given to 55,610 women, leading to 382 new cases of HMB, having an incidence rate of 686.93 cases per 100,000 inhabitants. Most cases were associated with Pfizer/BioNTech^®^, with 361 cases (IR = 666.0), and some with Moderna^®^, with 21 cases (IR = 1538.5) (Table 2).

For dose 5, of the 199 women who received this dose, less than five new cases of HMB were linked to Pfizer/BioNTech^®^ (IR = 1505.5) (Table 2).

#### 3.1.7. Association between COVID-19 Vaccination and Heavy Menstrual Bleeding by Types and Doses

The association of heavy menstrual bleeding from first to second COVID-19 vaccine dose by vaccine type. This analysis included 1,063,341 women to analyze the association between the new diagnoses of HMB during the time between the first and the second vaccine doses. The administration of the AstraZeneca^®^ vaccine was associated with an increase in the odds of developing HMB compared to the Pfizer/BioNTech^®^ vaccine, with an odds ratio (OR) of 1.77 (95% confidence interval [CI]: 1.53, 2.03). Meanwhile, the Janssen^®^ vaccine showed an OR of 2.16 (CI: 1.87, 2.48), indicating a significant increase in the odds of new HMB cases. In contrast, Moderna^®^ was not statistically significantly associated with HMB. Each additional day between the administration of the first and second doses slightly increased the odds of HMB, with an OR of 1.007 (CI: 1.007, 1.008). Regarding age, the groups of 36 to 45 years and 46 to 55 years showed significant increases in the odds of HMB, with ORs of 1.86 (CI: 1.50, 2.34) and 1.75 (CI: 1.41, 2.21), respectively. COVID-19 infection, hypertension, and obesity were associated with a significant increase in the odds of HMB, with ORs of 1.22 (CI: 1.11, 1.35), 1.70 (CI: 1.41, 2.03), and 1.37 (CI: 1.11, 1.68), respectively. Chronic respiratory diseases also showed an increase, with an OR of 1.40 (CI: 1.19, 1.63). The use of hormonal medication was notably associated with an increase in the odds of HMB, with an OR of 3.21 (CI: 2.88, 3.58) (Table 3).

The association of heavy menstrual bleeding from the second to the third COVID-19 vaccine dose by vaccine type. This analysis was based on 534,570 cases, focusing on the period between the second and third doses. The second doses of Janssen^®^ and the third doses of AstraZeneca^®^ were not included in the model. The administration of the Moderna^®^ vaccine was associated with a reduction in the odds of HMB, with an OR of 0.85 (CI: 0.77, 0.94). In contrast, the AstraZeneca^®^ vaccine was not statistically significantly associated with HMB. Each additional day between the second and third doses slightly increased the odds of developing HMB, with an OR of 1.004 (CI: 1.004, 1.004). Regarding age groups, the results showed that, compared to the reference group of 12–18 years, individuals aged 19 to 25 years had an OR of 0.75 (CI: 0.58, 0.97). On the other hand, the age groups of 36 to 45 years and 46 to 55 years showed increases in the odds of HMB, with ORs of 1.60 (CI: 1.29, 2.03) and 1.32 (CI: 1.05, 1.67), respectively. COVID-19 infection showed an increase in the odds of developing HMB, with an OR of 1.09 (CI: 1.02, 1.117). Also, hypertension and diabetes showed an increase in the odds of HMB, evidenced by an OR of 1.42 (CI: 1.24, 1.61) for hypertension and an OR of 1.33 (CI: 1.05, 1.65) for diabetes. Similarly, obesity and chronic respiratory diseases also showed increases in the odds of HMB, with ORs of 1.24 (CI: 1.06, 1.44) and 1.18 (CI: 1.05, 1.32), respectively. Serious heart diseases stood out with a significantly high OR of 1.74 (CI: 1.07, 2.65). Lastly, the administration of hormonal medication was associated with a marked increase in the odds of HMB, with an OR of 3.90 (CI: 3.61, 4.21) (Table 4).

### 3.2. Amenorrhea

#### 3.2.1. Socio-Demographic Characteristics

A significantly higher percentage of women experiencing AM (25.1%) were born outside of Spain, compared to those without AM (20.8%) (*p* < 0.001). The age of women with AM was significantly lower (28.38 years ± 11.28) than those without this condition (35.87 years ± 12.44). Similarly, differences in age distribution were statistically significant, with a higher proportion of younger women in the AM group (Table 1).

The results indicated statistically significant differences in AM prevalence across socioeconomic quintiles. The lower socioeconomic status quintiles, specifically quintiles 4 and 5, showed the highest prevalence of AM (22.3% and 23.8%, respectively) compared to their representation in the total population. Conversely, quintiles 1 and 2 had a lower prevalence of AM relative to the general population (15.3% vs. 21% and 18.5% vs. 20.5%, respectively) and compared to other quintiles (*p* < 0.001) (Table 1).

#### 3.2.2. Chronic Degenerative Diseases

Women with AM exhibited a higher prevalence of various medical conditions. Type 1 and type 2 diabetes were found in 1.0% of the group with AM, compared to 1.2% in the group without this disorder (*p* = 0.001). Auto-immune diseases (1.1% vs. 0.8%; *p* < 0.001) and polycystic ovary syndrome (2.4% vs. 0.9%; *p* < 0.001) showed higher prevalence in the group with AM. On the contrary, hypertension was more prevalent among women without AM (2.3% vs. 3.8%; *p* < 0.001) (Table 1).

Furthermore, chronic respiratory diseases were present in 8.0% of women with AM, compared to 6.5% of those without it (*p* < 0.001). Chronic kidney diseases showed statistically significant differences, albeit with low prevalence (0.2% in women with AM vs. 0.3% in the group without AM; *p* = 0.028). Diseases related to metabolic disorders showed differences. Dyslipidemia was also more frequent in the group without AM (2.1% vs. 4.7%). On the contrary, obesity was more frequent in the group with AM (4.7% vs. 3.6%) (Table 1).

#### 3.2.3. Hormonal Medication Use

The hormonal medication used was significantly higher in the group with AM compared to the group without AM (*p* < 0.001). (28.4% vs. 11.0%, *p* < 0.001) (Table 1).

#### 3.2.4. Lifestyle

No significant differences were observed (Table 1).

#### 3.2.5. History of COVID-19

A higher percentage of women in the group with AM (52.8%) had been infected with COVID-19 compared to those without AM (49.7%) (*p* < 0.001) (Table 1).

#### 3.2.6. Vaccination Status and Incidence Rate of Amenorrhea

Differences in the number of vaccine doses received between the groups were minor, with the majority receiving between two and three doses (Table 1).

For dose 1, out of 1,172,621 vaccinated women, there were 6542 new cases of amenorrhea, resulting in an incidence rate of 557.9 cases per 100,000 inhabitants. Pfizer/BioNTech^®^ had the majority with 3661 cases (IR = 446.9), followed by Moderna^®^, with 1540 cases (IR = 743.8), Janssen^®^, with 1039 cases (IR = 1623.3), and AstraZeneca^®^, with 302 cases (IR = 366.4) (Table 2).

For dose 2, Among 1,063,341 vaccinated women, there were 23,311 new cases of AM, having an incidence rate of 2192.2 cases per 100,000 inhabitants. Pfizer/BioNTech^®^ had the highest number of cases with 17,298 cases (IR = 2302.2), then Moderna^®^, with 5235 cases (IR = 2169.7), and AstraZeneca^®^, with 773 cases (IR = 1095.2). Janssen^®^ had ≤ 5 cases (Table 2).

For dose 3, a total of 534,621 women received this dose, from which 8105 new cases of AM were reported, giving an incidence rate of 1516.0 cases per 100,000 inhabitants. Moderna^®^ accounted for 6200 cases (IR = 1458.8) and Pfizer/BioNTech^®^ for 1902 cases (IR = 1736.6). AstraZeneca^®^ had ≤ 5 cases (Table 2).

For dose 4, given to 55,610 women, it resulted in 333 new cases of AM, resulting in an incidence rate of 598.8 cases per 100,000 inhabitants. The majority were from Pfizer/BioNTech^®^, with 313 cases (IR = 557.4), and some from Moderna^®^, with 20 cases (IR = 1428.6) (Table 2).

For dose 5, of the 199 women who received this dose, there were no new cases of amenorrhea (Table 2).

#### 3.2.7. Association between COVID-19 Vaccination and Amenorrhea by Types and Doses

The association of amenorrhea from the first to the second COVID-19 vaccine dose by vaccine type. The analysis results show significant variations in the odds of AM associated with different vaccines compared to Pfizer/BioNTech^®^. AstraZeneca^®^ was associated with a significant increase in the odds of developing AM, with an OR of 1.62 (CI: 1.42, 1.85). Janssen^®^ also showed a notable increase, with an OR of 1.99 (CI: 1.74, 2.27), indicating a substantial increase in the probability of developing AM. In contrast, the Moderna^®^ vaccine was not statistically significantly associated with AM. Regarding the time interval between the first and second doses, each additional day was associated with a slight increase in the odds of developing AM, with an OR of 1.007 (CI: 1.007, 1.008). Age was a significant factor in the probability of developing AM. Younger groups, particularly those aged 19 to 25 and 26 to 35, compared to the reference group of 12 to 18 years, showed ORs of 1.01 (CI: 0.87, 1.17) and 0.72 (CI: 0.62, 0.83), respectively. Older groups, aged 36 to 45 and 46 to 55, showed ORs of 0.50 (CI: 0.43, 0.58) and 0.24 (CI: 0.20, 0.29), respectively. COVID-19 infection showed an increase in the odds of developing AM, with an OR of 1.13 (CI: 1.03, 1.23). Diabetes mellitus showed an increase in the odds, with an OR of 1.62 (CI: 1.14, 2.23), and chronic respiratory diseases increased the odds, with an OR of 1.25 (CI: 1.07, 1.45). Auto-immune diseases were also significant, with an OR of 1.63 (CI: 1.10, 2.32). Finally, the administration of hormonal medication was associated with a marked increase in the odds of AM, with an OR of 2.91 (CI: 2.65, 3.19).

The association of amenorrhea from the second to the third COVID-19 vaccine dose by vaccine type. The analysis results indicate significant variations in the odds of developing AM associated with different vaccines compared to Pfizer/BioNTech^®^. The Moderna^®^ vaccine showed an OR of 0.86 (CI: 0.79, 0.94). On the other hand, the AstraZeneca^®^ vaccine was not statistically significantly associated with AM. The time interval between the second and third doses showed a slight increase in the odds of developing AM for each additional day, with an OR of 1.004 (CI: 1.004, 1.004), highlighting a cumulative effect of the interval between doses. Regarding age groups, it was observed that, compared to the reference group of 12–18 years, the age groups of 19 to 25 years and 26 to 35 years experienced ORs of 0.72 (CI: 0.63, 0.82) and 0.45 (CI: 0.40, 0.52), respectively. The older age groups of 36 to 45 years and 46 to 55 years showed ORs of 0.38 (CI: 0.34, 0.43) and 0.16 (CI: 0.14, 0.18). Auto-immune diseases were associated with an increase in the odds of AM, with an OR of 1.52 (CI: 1.14, 1.99). Obesity also showed a significant association, with an OR of 1.32 (CI: 1.14, 1.52). Additionally, polycystic ovary syndrome was related to an increase in the odds of AM, with an OR of 1.75 (CI: 1.46, 2.09). Hormonal medication use was notably associated with an increase in the odds of AM, with an OR of 2.90 (CI: 2.70, 3.10).

## 4. Discussion

In the present study, we found an increase in the incidence of both HMB and AM in the second and third doses of vaccines relative to the other doses, a pattern that aligns with observations from other studies. For example, in Japan, Hosoya et al. reported a significant prolongation of the menstrual cycle following booster doses, suggesting a dose-dependent increase in menstrual disorders [24]. Similarly, Namiki et al. found that abnormal bleeding and irregular menstrual cycles were more common after the third dose compared to the first [25].

The first dose of the Janssen^®^ vaccine, administered as a single-dose regimen, exhibited a higher incidence of menstrual disorders in our study. This contrasts with multi-dose vaccines, where such incidences were less prominent initially but increased with subsequent doses. The extended interval before administering a second booster dose for Janssen^®^ recipients may account for this disparity, which is similar to those who received the AstraZeneca^®^ booster doses. These cases are also residual, rendering the reported incidences potentially unrepresentative.

Our analysis reveals varied associations with COVID-19 vaccine types, number of doses, and other covariates, underscoring the complex nature of vaccine-induced reactions. We found that the administration of the AstraZeneca^®^ and Janssen^®^ vaccines was associated with a higher probability of diagnosing HMB and AM during the period from the first to the second dose when compared to Pfizer/BioNTech^®^. Conversely, Moderna^®^ demonstrated a protective effect against these conditions in the period between the second and third doses. This protective role of Moderna^®^ contrasts with findings from Lessans et al., who reported high rates of irregular bleeding and menstrual changes following Pfizer/BioNTech^®^ vaccination but did not observe a similar pattern with Moderna^®^ [26]. Additionally, Alahmadi et al. reported that the Moderna^®^ vaccine was significantly associated with the highest rate of menstrual changes, while AstraZeneca^®^ had the lowest rate [11].

This discrepancy in results compared with other studies underscores the need for further investigation into the differential impacts of vaccine types on menstrual health. Furthermore, the lack of established background incidence rates for menstrual disorders presents a significant limitation in comparing our findings to a baseline [27,28]. The absence of these rates impedes our ability to fully contextualize the observed increase in HMB and AM in our study. A meta-analysis conducted by Al Kadri et al. (2023) demonstrated significant heterogeneity among studies, suggesting that the impact of vaccination on menstrual health varies widely across different populations and geographical regions [20]. This variability reinforces the necessity for ongoing monitoring and thorough analysis to understand the extent of vaccine-induced menstrual changes.

Moreover, our findings suggest that various factors, including underlying comorbidities and chronic medication use, are significant determinants that influence changes in menstrual parameters. Insights from Klein et al. elucidate how biological sex disparities contribute to divergent COVID-19 infection outcomes, potentially heightening the susceptibility to menstrual disorders following immunization [29]. Most physiological processes in the female reproductive system are intertwined with inflammatory responses. These inflammatory mediators are critical for various functions, including tissue repair, angiogenesis, and the cyclic degradation, remodeling, and proliferation of the endometrial lining [30]. Central to these processes are cytokines and chemokines, which serve as essential regulators within the uterine environment and are active throughout the menstrual cycle [31].

COVID-19 infection is recognized as a pro-inflammatory condition that is known to trigger a cytokine storm, leading to profound immune exhaustion. It has been observed that SARS-CoV-2 infection can disrupt the menstrual cycle, affecting women irrespective of their vaccination status, thus underscoring the extensive impact of inflammatory responses on reproductive health dynamics [32].

The interaction between the immune system and hormonal regulation is pivotal in understanding a possible explanation for menstrual changes post-vaccination. The vaccine’s impact on the hypothalamic–pituitary–ovarian axis, as noted by Minakshi et al., may explain the increased incidence of menstrual disorders observed [33]. The pandemic’s role in exacerbating stress and systemic inflammation could further influence these effects, hypothalamic hypogonadism, which results in menstrual irregularities, as suggested by studies on the physiological impacts of psychological stress [34].

Our analysis underscores an association of hormonal medications on menstrual disorders post-vaccination. On the contrary, the observed protective effects of estradiol-containing contraceptives, as reported by Alvergne et al. (2023), indicate promising research directions for mitigating the adverse effects of vaccines through hormonal treatments [35].

Both AM and HMB have been found to be more prevalent in lower socioeconomic status, although there are limitations in the overall availability of these data. One hypothesis that could corroborate this information is that conditions in the highest quintiles may be underestimated due to the potential utilization of private gynecological healthcare, thus not fully reflected in our database. Conversely, this may reinforce the findings of Medina-Perucha et al., wherein experiences of financial hardship and poorer self-rated health may influence or mediate menstrual characteristics [36].

This study has several strengths. First, the use of the SIDIAP database covers a broad demographic in Catalonia, ensuring that our findings are representative and potentially applicable to similar populations, which might not be feasible with smaller datasets. This data supports robust statistical analysis and enables the identification of associations between vaccination and menstrual disorders. Additionally, the comprehensive data used in the study included clinical, sociodemographic, lifestyle, and pharmaceutical covariates, allowing for a thorough evaluation of factors that may influence menstrual disorders. The interval between doses was calculated and included in the analysis and emerged as a significant factor in our study. Timing can affect the biological response to vaccine administration and impact menstrual health. This extensive dataset strengthens our analysis and aids in understanding the complex interactions between health factors and post-vaccination side effects. The database also meticulously records all COVID-19 vaccinations, including dosage and manufacturer details, providing a deeper insight into the impact of different vaccination regimes on menstrual health.

Despite the many strengths of this study, several limitations must be acknowledged. First, as a retrospective cohort observational study, the establishment of causality cannot be made. Additionally, although multiple covariates were included to adjust the analyses, some important confounding factors may not have been fully controlled. Another significant limitation is the potential underestimation or overestimation of the events of interest due to reliance on EHRs for identifying cases of HMB and AM. The accuracy of these records depends on the consistency of healthcare documentation, with variability introducing misclassification bias and affecting reliability. There is also a possibility that some women identified as not having menstrual disorders might actually have had them, but these were not recorded in the database. This could introduce a potential source of bias, as the SIDIAP database may not capture all instances of menstrual disorders if these were undiagnosed, unreported by patients, or unrecorded by healthcare providers outside of the public healthcare system, potentially skewing our conclusions.

Furthermore, our analysis did not exclude women who had undergone hysterectomies. Moreover, data for sex was taken from information in the registry rather than from patient-reported gender, which may not accurately reflect gender identity and could affect the interpretation of our findings.

## 5. Conclusions

This study provides a detailed analysis of the associations between COVID-19 vaccination and conditions such as heavy menstrual bleeding and amenorrhea, examining a large cohort of over 1.17 million vaccinated women aged 12 to 55 years. The findings highlight how different vaccine types and doses influence menstrual health, identifying the interval between doses as a critical factor in the onset of menstrual disorders.

The study emphasizes the complexity of vaccine-induced reactions and the importance of rigorous monitoring of menstrual health post-vaccination. It also highlights the need for standardized menstrual parameters and enhanced data collection to better understand the impacts of vaccines. Despite limitations such as potential underreporting and database constraints, this study provides valuable insights for shaping public health strategies and clinical guidelines.

Furthermore, it underscores the importance of incorporating menstrual health considerations into clinical trials for the development of new vaccines and pharmaceuticals, ensuring systematic evaluations that can improve healthcare decisions and patient outcomes. Ongoing research is crucial to validate these findings and to refine public health strategies and clinical guidelines. Future studies should expand the data sources used and address the identified limitations to provide a more comprehensive understanding of the vaccine’s impact on menstrual health.

## Figures and Tables

**Figure 1 ijerph-21-01090-f001:**
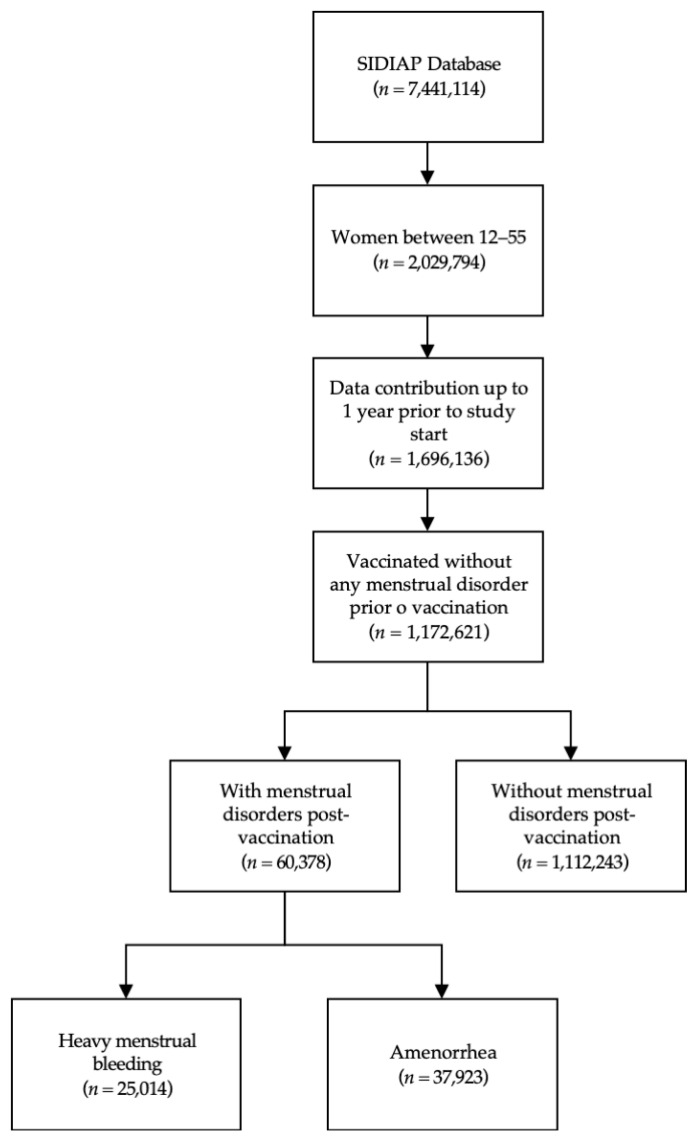
Study population flowchart.

**Table 1 ijerph-21-01090-t001:** Characteristics of COVID-19 vaccinated women included in the study cohort.

	Heavy Menstrual Bleeding			Amenorrhea		
	yes	no		yes	no	
	(*n* = 25,014)	(*n* = 1,147,607)		(*n* = 37,923)	(*n* = 1,134,698)	
Socio-demographic and socioeconomic characteristics						
Country of Birth			***			***
Out of Spain	5266 (26.5)	182,041 (20.8)		7574 (25.1)	179,733 (20.8)	
Spain	14,597 (73.5)	691,697 (79.2)		22,654 (74.9)	683,640 (79.2)	
Missing	5151	273,869		7695	271,325	
Age (in years)			***			***
Mean (SD)	36.50 (11.25)	35.61 (12.50)		28.38 (11.28)	35.87 (12.44)	
Median	39.60	37.40		26.70	37.80	
Q1–Q3	28.70–45.30	25.30–46.30		18.00–38.50	25.80–46.40	
Min–Max	12.00–55.00	12.00–55.00		12.00–55.00	12.00–55.00	
Age (by groups)			***			***
12–18	2562 (10.2)	143,783 (12.5)		9289 (24.5)	137,056 (12.1)	
19–25	2254 (9.0)	134,618 (11.7)		7975 (21.0)	128,897 (11.4)	
26–35	4416 (17.7)	231,483 (20.2)		8464 (22.3)	227,435 (20.0)	
36–45	9069 (36.3)	307,643 (26.8)		8801 (23.2)	307,911 (27.1)	
46–55	6713 (26.8)	330,080 (28.8)		3394 (8.9)	333,399 (29.4)	
Socioeconomic index MEDEA			***			***
Q1	627 (14.4)	40,723 (21.0)		982 (15.3)	40,368 (21.0)	
Q2	857 (19.7)	39,770 (20.5)		1188 (18.5)	39,439 (20.5)	
Q3	956 (22.0)	39,335 (20.3)		1287 (20.1)	39,004 (20.3)	
Q4	967 (22.2)	38,183 (19.7)		1430 (22.3)	37,720 (19.6)	
Q5	941 (21.6)	36,064 (18.6)		1529 (23.8)	35,476 (18.5)	
Missing	20,666	953,532		31,507	942,691	
Concomitant chronic-degenerative diseases						
Type 1 and 2 Diabetes Mellitus			***			***
Yes	445 (1.8)	13,659 (1.2)		389 (1.0)	13,715 (1.2)	
No	24,569 (98.2)	1,133,948 (98.8)		37,534 (99.0)	1,120,983 (98.8)	
HIV			*			*
Yes	4 (0.0)	255 (0.0)		4 (0.0)	255 (0.0)	
No	25,010 (100.0)	1,147,352 (100.0)		37,919 (100.0)	1,134,443 (100.0)	
Hypertension			***			***
Yes	1418 (5.7)	42,634 (3.7)		858 (2.3)	43,194 (3.8)	
No	23,596 (94.3)	1,104,973 (96.3)		37,065 (97.7)	1,091,504 (96.2)	
Auto-immune chronic diseases			***			***
Yes	279 (1.1)	8948 (0.8)		404 (1.1)	8823 (0.8)	
No	24,735 (98.9)	1,138,659 (99.2)		37,519 (98.9)	1,125,875 (99.2)	
Chronic kidney disease			***			**
Yes	110 (0.4)	3518 (0.3)		94 (0.2)	3534 (0.3)	
No	24,904 (99.6)	1,144,089 (99.7)		37,829 (99.8)	1,131,164 (99.7)	
Dyslipidemia			***			***
Yes	844 (3.4)	31,765 (2.8)		789 (2.1)	31,820 (2.8)	
No	24,170 (96.6)	1,115,842 (97.2)		37,134 (97.9)	1,102,878 (97.2)	
Obesity			***			***
Yes	1225 (4.9)	41,146 (3.6)		1792 (4.7)	40,579 (3.6)	
No	23,789 (95.1)	1,106,461 (96.4)		36,131 (95.3)	1,094,119 (96.4)	
Polycystic ovary syndrome			***			***
Yes	339 (1.4)	10,290 (0.9)		925 (2.4)	9704 (0.9)	
No	24,675 (98.6)	1,137,317 (99.1)		36,998 (97.6)	1,124,994 (99.1)	
Chronic respiratory diseases			***			***
Yes	2302 (9.2)	74,492 (6.5)		3021 (8.0)	73,773 (6.5)	
No	22,712 (90.8)	1,073,115 (93.5)		34,902 (92.0)	1,060,925 (93.5)	
Serious heart conditions			***			*
Yes	89 (0.4)	2327 (0.2)		69 (0.2)	2347 (0.2)	
No	24,925 (99.6)	1,145,280 (99.8)		37,854 (99.8)	1,132,351 (99.8)	
Hormonal medication						
Sex hormones and modulators			***			***
Yes	7038 (28.1)	128,314 (11.2)		10,753 (28.4)	124,599 (11.0)	
No	17,976 (71.9)	1,019,293 (88.8)		27,170 (71.6)	1,010,099 (89.0)	
Lifestyle						
Alcohol consumption			*			*
Nonalcoholic	3385 (56.8)	157,030 (57.9)		5136 (58.2)	155,279 (57.9)	
Low Risk	2478 (41.6)	109,386 (40.4)		3547 (40.2)	108,317 (40.4)	
High Risk	100 (1.7)	4582 (1.7)		138 (1.6)	4544 (1.7)	
Missing	19,051	876,609		29,102	866,558	
IMC			*			*
Mean (SD)	27.49 (5.47)	27.48 (5.61)		27.46 (5.54)	27.48 (5.61)	
Median	27.07	26.96		26.98	26.96	
Q1–Q3	23.81–30.59	23.68, 30.66		23.76, 30.47	23.68, 30.67	
Min–Max	14.22–54.80	6.3859.82		11.6058.48	6.3859.82	
Missing	20,842	959,013		31,854	948,001	
Smoking status			*			*
Non-smoker	3901 (72.0)	178,055 (72.4)		5822 (72.6)	176,134 (72.4)	
Smoker	1011 (18.7)	46,436 (18.9)		1494 (18.6)	45,953 (18.9)	
Former smoker	503 (9.3)	21,388 (8.7)		708 (8.8)	21,183 (8.7)	
Missing	19,599	901,728		29,899	891,428	
Sleeping duration (daily)			*			*
Mean (SD)	7.50 (2.12)	7.19 (1.58)		6.83 (1.40)	7.21 (1.59)	
Median	7.50	7.00		6.50	7.00	
Q1–Q3	6.75–8.25	6.00–8.00		6.00–8.00	6.00, 8.00	
Min–Max	6.00–9.00	3.00–18.00		5.00–9.00	3.0018.00	
Missing	25,012	1,147,355		37,911	1,134,456	
COVID-19 history						
COVID-19 vaccination (dose)			**			*
Mean (SD)	2.40 (0.73)	2.41 (0.72)		2.26 (0.69)	2.42 (0.72)	
Median	2.00	2.00		2.00	2.00	
Q1–Q3	2.00–3.00	2.00–3.00		2.00–3.00	2.00–3.00	
Min–Max	1.00–5.00	1.00–5.00		1.00–5.00	1.00–5.00	
COVID-19 infection			***			***
Yes	13,520 (54.0)	569,949 (49.7)		20,009 (52.8)	563,460 (49.7)	
No	11,494 (46.0)	577,658 (50.3)		17,914 (47.2)	571,238 (50.3)	

Data are number (percentage) unless otherwise stated. *** *p*-value: <0.001; ** *p*-value: <0.05; * *p*-value > 0.05.

**Table 2 ijerph-21-01090-t002:** Descriptive analysis and incidence of COVID-19 vaccination and heavy menstrual bleeding and amenorrhea by dose and vaccine type.

	Heavy Menstrual Bleeding				Amenorrhea			
	N (%)		*p*-Value	IR *	N (%)		*p*-Value	IR *
Dose 1 by manufacturer (*n* = 1,172,621)								
AstraZeneca^®^	272	(6.6)	<0.001	330.0	302	(4.6)	<0.001	366.4
Janssen^®^	914	(22.0)		1428.0	1039	(15.9)		1623.3
Moderna^®^	883	(21.3)		426.5	1540	(23.5)		743.8
Pfizer/BioNTech^®^	2083	(50.2)		254.3	3661	(56.0)		446.9
Total	4152			354.1	6542			557.9
Dose 2 by manufacturer (*n* = 1,063,341)								
AstraZeneca^®^	625	(4.5)	<0.001	885.5	773	(3.3)	<0.001	1095.2
Janssen^®^	<5	(0.0)		1851.8	5	(0.0)		4629.6
Moderna^®^	3261	(23.5)		1351.6	5235	(22.5)		2169.7
Pfizer/BioNTech^®^	9987	(72.0)		1329.1	17,298	(74.2)		2302.2
Total	13,875			1304.8	23,311			2192.2
Dose 3 by manufacturer (*n* = 534,621)								
AstraZeneca^®^	<5	(0.0)	<0.001	1515.1	<5	(0.0)	<0.001	4545.5
Janssen^®^	0	(0.0)		0	0	(0.0)		0.0
Moderna^®^	5828	(83.8)		1371.24	6200	(76.5)		1458.8
Pfizer/BioNTech^®^	1125	(16.2)		1027.15	1902	(23.5)		1736.6
Total	6954			1300.73	8105			1516.0
Dose 4 by manufacturer (*n* = 55,610)								
AstraZeneca^®^	0	(0.0)	0.003	0.0	0	(0)	<0.001	0.0
Janssen^®^	0	(0.0)		0.9	0	(0)		0.0
Moderna^®^	21	(5.)		1500.0	20	(6.0)		1428.6
Pfizer/BioNTech^®^	361	(94.5)		666.0	313	(94.0)		577.4
Total	382			686.93	333			598.8
Dose 5 by manufacturer (*n* = 199)								
AstraZeneca^®^	0	(0.0)	0.803	0.0	0	(0.0)		0.0
Janssen^®^	0	(0.0)		0.0	0	(0.0)		0.0
Moderna^®^	0	(0.0)		0.0	0	(0.0)		0.0
Pfizer/BioNTech^®^	<5	(100.0)		1538.5	0	(0.0)		0.0
Total	<5			1505.5	0			0.0

Data are number (percentage) unless otherwise stated. * IR = Incidence rate per 100,000 individuals.

**Table 3 ijerph-21-01090-t003:** Association between COVID-19 vaccine doses and HMB and AM between first and second dose. (Adjusted model).

	Heavy Menstrual Bleeding		Amenorrhea	
	OR	95% CI	OR	95% CI
Pfizer/BioNTech^®^ (Ref.)				
AstraZeneca^®^	1.765	(1.527–2.033)	1.623	(1.416–1.854)
Janssen^®^	2.155	(1.873–2.476)	1.989	(1.740–2.269)
Moderna^®^	1.153	(0.997–1.329)	0.992	(0.878–1.119)
Time between first and second dose	1.007	(1.007–1.007)	1.007	(1.007–1.008)
Age 12–18 (Ref.)				
Age 19–25	0.979	(0.755–1.274)	1.011	(0.873–1.173)
Age 26–35	1.105	(0.875–1.408)	0.718	(0.622–0.829)
Age 36–45	1.863	(1.500–2.340)	0.498	(0.431–0.577)
Age 46–55	1.752	(1.406–2.206)	0.241	(0.202–0.287)
COVID-19 infection	1.222	(1.106–1.351)	1.132	(1.035–1.238)
Type 1 and 2 Diabetes Mellitus	1.027	(0.712–1.430)	1.616	(1.135–2.228)
Hypertension	1.699	(1.413–2.028)	0.831	(0.617–1.095)
Auto-immune chronic diseases	1.476	(0.946–2.181)	1.634	(1.101–2.324)
Chronic kidney disease	0.694	(0.273–1.430)	1.281	(0.542–2.532)
Dyslipidemia	1.156	(0.906–1.452)	1.409	(1.077–1.811)
Obesity	1.376	(1.115–1.679)	1.512	(1.249–1.813)
Polycystic ovary syndrome	1.456	(0.973–2.086)	1.962	(1.525–2.482)
Chronic respiratory diseases	1.396	(1.190–1.629)	1.247	(1.068–1.447)
Serious heart conditions	1.403	(0.598–2.750)	1.056	(0.326–2.479)
Sex hormones and modulators	3.214	(2.879–3.583)	2.908	(2.649–3.189)

Total number of observations: 1,063,341. Data are numbers. OR: odds ratio; 95% CI: 95% confidence interval.

**Table 4 ijerph-21-01090-t004:** Association between COVID-19 vaccine doses and HMB and AM between second and third dose. (Adjusted model).

	Heavy Menstrual Bleeding		Amenorrhea	
	OR	95% CI	OR	95% CI
Pfizer/BioNTech^®^ (Ref.)				
AstraZeneca^®^	1.050	(0.935–1.175)	0.950	(0.852–1.057)
Moderna^®^	0.852	(0.771–0.939)	0.862	(0.791–0.938)
Time between second and third dose	1.004	(1.004–1.004)	1.004	(1.004–1.004)
Age 12–18 (Ref.)				
Age 19–25	0.745	(0.580–0.968)	0.719	(0.631–0.821)
Age 26–35	0.967	(0.768–1.233)	0.454	(0.400–0.516)
Age 36–45	1.605	(1.287–2.031)	0.380	(0.336–0.432)
Age 46–55	1.316	(1.054–1.667)	0.160	(0.139–0.183)
COVID-19 infection	1.088	(1.016–1.166)	1.023	(0.961–1.089)
Type 1 and 2 Diabetes Mellitus	1.327	(1.052–1.648)	1.276	(0.973–1.639)
Hypertension	1.420	(1.244–1.614)	1.122	(0.943–1.325)
Auto-immune chronic diseases	1.272	(0.914–1.717)	1.524	(1.140–1.988)
Chronic kidney disease	1.363	(0.887–1.994)	0.814	(0.420–1.408)
Dyslipidemia	1.108	(0.938–1.300)	1.180	(0.976–1.414)
Obesity	1.240	(1.061–1.439)	1.320	(1.138–1.522)
Polycystic ovary syndrome	0.973	(0.703–1.308)	1.752	(1.457–2.088)
Chronic respiratory diseases	1.180	(1.049–1.323)	1.098	(0.980–1.226)
Serious heart conditions	1.740	(1.074–2.652)	0.649	(0.256–1.326)
Sex hormones and modulators	3.898	(3.611–4.205)	2.896	(2.704–3.101)

Total number of observations: 534,570. Data are numbers. OR: odds ratio; 95% CI: 95% confidence interval.

## Data Availability

The data are readily available upon request.

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
