# Peer review of "Association between COVID-19 Vaccines and Menstrual Disorders: Retrospective Cohort Study of Women Aged 12–55 Years Old in Catalonia, Spain"

_ijerph, 2024, doi:10.3390/ijerph21081090_

Round 1

Reviewer 1 Report

Comments and Suggestions for Authors

Thank you very much for allowing me to review this paper. This study presents results that provide further evidence of a potential adverse effect associated with COVID-19 vaccination, which has already been reported in other studies with different designs. Congratulations to the authors for this hard work analyzing a population-based database as presented. There are few studies that analyze a purely population-based sample. However, I would like the authors to clarify several questions that arose while reading this paper.

  1. Title: Clear and concise, written based on the main objective.
  2. Abstract: Well-structured. I would like the authors to clarify whether the OR results they present are crude ORs or ORs adjusted based on covariates that could act as confounders in the studied effect.
  3. Introduction: From my point of view, this section should be improved. The introduction does not seem well-structured. I believe it should be structured into three clearly defined paragraphs:
    • First paragraph: Importance of the topic: Here, the what and why of the work should be specified with current and novel bibliographic support, as well as the academic value that, as a justification, captivates the reader. Then continue with the conceptual or historical background of the topic: The goal is to temporally and spatially contextualize the reader through a brief overview of the past and present of the topic, to justify how it will be addressed and as a strategy to arouse academic interest in the article.
    • Second paragraph: Definition of the problem: After explaining the general problem, address the nature and scope of the research problem. This should be presented concisely, stated as a question that has not been answered but which the article attempts to resolve. The magnitude and importance of the problem should be highlighted to consider it for the development of new projects.
    • Third paragraph: Definition of the objective: This should align with the research question and be consistent with the rest of the article. This allows the reader to contextualize the theoretical framework of the study. On the other hand, the last two paragraphs of the current introduction are more parts of the methodology or discussion, not of the introduction itself.
  4. Methodology: a. I have a serious doubt whether the study design is really a retrospective study as the authors state, or a prospective follow-up study of a historical cohort. In my opinion, the design of this work fits the design of a cohort follow-up study. Women vaccinated without menstrual abnormalities are selected and followed prospectively, knowing the moments they are vaccinated with each COVID-19 vaccine dose, and knowing the moment when the outcome variable occurs. Therefore, I would reconsider the design if this is the case. b. In section 2.2 Population, when referring to the registered trademarks of the vaccines, the registered trademark symbol should be included. c. In the statistical analysis, it is stated that the incidence of HMB and AM events is calculated. However, this is not reported in the results. On the other hand, if the incidence can be calculated, it is because the design is prospective; otherwise, incidence could never be calculated. d. Is the database used sufficiently robust? It is possible that patients selected as not having menstrual abnormalities actually had them, but they were not recorded in the database?
  5. Results: a. 2560 women presented both conditions under study. How were these women treated? Were they excluded from the analysis? b. The mean age was 35.61 ± 12.50. It seems It seems there is significant variability in age. Were diagnoses of perimenopausal states taken into account?

    c. The "incidence rate" is reported in section 3.1.6. Was the incidence rate actually calculated? From my point of view, there is confusion regarding the calculation of incidence in this study. What calculation was performed? Incidence rate? Cumulative incidence? This needs to be clarified. Additionally, neither of these should be presented as a percentage in the context of this study.

    d. On the other hand, incidence calculations are performed, indicating a prospective temporality, but a logistic regression is conducted. Why not perform a Cox proportional hazards model and calculate the hazard ratio? These results would be more robust.

    e. It is also not clear in the results whether a risk measure (OR in this case) was calculated, adjusted for comorbidities and sociodemographic variables that could affect menstrual alteration. A table with the crude OR and adjusted OR values should be presented.

    1. Discussion: This section should be revised based on the clarifications of the previous doubts.

    Thank you very much.

Author Response

Thank you very much for allowing me to review this paper. This study presents results that provide further evidence of a potential adverse effect associated with COVID-19 vaccination, which has already been reported in other studies with different designs. Congratulations to the authors for this hard work analyzing a population-based database as presented. There are few studies that analyze a purely population-based sample. However, I would like the authors to clarify several questions that arose while reading this paper.

Comment 1.Title: Clear and concise, written based on the main objective.

Comment 2. Abstract: Well-structured. I would like the authors to clarify whether the OR results they present are crude ORs or ORs adjusted based on covariates that could act as confounders in the studied effect.

Response 2: Thank you for your comment. We have, accordingly, revised the abstract to clarify that the OR results presented are adjusted ORs. These adjustments were made for potential confounders, including age, prior health conditions, and other relevant covariates. This change can be found in the revised manuscript on page 1, lines 20-27.

Comment 3. Introduction: From my point of view, this section should be improved. The introduction does not seem well-structured. I believe it should be structured into three clearly defined paragraphs:

  • First paragraph: Importance of the topic: Here, the what and why of the work should be specified with current and novel bibliographic support, as well as the academic value that, as a justification, captivates the reader. Then continue with the conceptual or historical background of the topic: The goal is to temporally and spatially contextualize the reader through a brief overview of the past and present of the topic, to justify how it will be addressed and as a strategy to arouse academic interest in the article.
  • Second paragraph: Definition of the problem: After explaining the general problem, address the nature and scope of the research problem. This should be presented concisely, stated as a question that has not been answered but which the article attempts to resolve. The magnitude and importance of the problem should be highlighted to consider it for the development of new projects.
  • Third paragraph: Definition of the objective: This should align with the research question and be consistent with the rest of the article. This allows the reader to contextualize the theoretical framework of the study. On the other hand, the last two paragraphs of the current introduction are more parts of the methodology or discussion, not of the introduction itself.

Response 3: Agree. We have, accordingly, revised the introduction to improve its structure and clarity. The introduction is now divided into three clearly defined paragraphs, as suggested:

  1. First Paragraph: Importance of the Topic: We have highlighted the significance of the study, the context of the COVID-19 pandemic, and the potential implications of the research, supported by current and novel bibliographic references.
  2. Second Paragraph: Definition of the Problem: We have defined the research problem, emphasizing the nature and scope of the issue, and presenting it as an unresolved question.
  3. Third Paragraph: Definition of the Objective: We have stated the research objective clearly, aligning it with the research question and ensuring consistency with the rest of the article.

These changes can be found in the revised manuscript on page 2, lines 29-75.

Comment 4: Methodology:

  • Comment 4.1: I have a serious doubt whether the study design is really a retrospective study as the authors state, or a prospective follow-up study of a historical cohort. In my opinion, the design of this work fits the design of a cohort follow-up study. Women vaccinated without menstrual abnormalities are selected and followed prospectively, knowing the moments they are vaccinated with each COVID-19 vaccine dose, and knowing the moment when the outcome variable occurs. Therefore, I would reconsider the design if this is the case.
  • Response 4.1: Thanks for your comment. We do agree with your observations, a prospective follow-up study of a historical cohort is a type of observational study that combines elements of both retrospective and prospective study designs. We consider this study as a retrospective cohort study using existing data already collected and available in the SIDIAP database. We identify a cohort of women aged 12-55 years old without previous menstrual disorders, who have been exposed to a COVID-19 vaccine in the past and follow them through historical data to determine the outcome of interest. The start of the study period was on December 27, 2020, the date when the first dose of COVID-19 vaccine was identified in SIDIAP database, and the end of the study period is June 30th, 2023, the last data available in SIDIAP database.
  • Comment 4.2: In section 2.2 Population, when referring to the registered trademarks of the vaccines, the registered trademark symbol should be included.
  • Response 4.2: Thank you for pointing this out. We agree with this comment. Therefore, we have included the registered trademark symbols (®) for the vaccines mentioned.
  • Comment 4.3: In the statistical analysis, it is stated that the incidence of HMB and AM events is calculated. However, this is not reported in the results. On the other hand, if the incidence can be calculated, it is because the design is prospective; otherwise, incidence could never be calculated.
  • Response 4.3: Thank you for pointing this out. We agree with this comment. Therefore, we would like to clarify that the incidences of HMB and AM were calculated retrospectively in the selected cohort that included women without previous menstrual disorders. We calculated these incidences of new cases only for the period between vaccine doses. This change and clarification have been added to the manuscript. This can be found in the revised manuscript on page 3, lines 96-210.
  • Comment 4.4: Is the database used sufficiently robust? It is possible that patients selected as not having menstrual abnormalities actually had them, but they were not recorded in the database?
  • Response 4.d: Thank you for this important observation. Yes, this could indeed be a limitation of our study. The possibility that some patients may have had undiagnosed or unrecorded menstrual abnormalities is a potential source of bias. We acknowledge that the SIDIAP database, while comprehensive, may not capture all instances of menstrual abnormalities, particularly if they were not reported by the patients or recorded by healthcare providers. SIDIAP database has many quality processes and it is a potential database for pharmacoepidemiology studies, and it is register in the https://catalogues.ema.europa.eu/node/1019/administrative-details. We have added a statement to the limitations section of the manuscript to clarify this potential bias. This change can be found in the revised manuscript on page 10, lines 663-675.

Comment 5: Results

  • Comment 5.1: 2560 women presented both conditions under study. How were these women treated? Were they excluded from the analysis?
  • Response 5.1: Thank you for your insightful question. We agree that the treatment of women diagnosed with both conditions requires clarification. Women who were diagnosed with both heavy menstrual bleeding (HMB) and amenorrhea (AM) were not excluded from the analysis. Instead, they were included in both groups for the respective analyses of HMB and AM. This approach allowed us to comprehensively assess the associations between COVID-19 vaccination and each condition independently. We have updated the manuscript to specify this clearly. This change can be found in the revised manuscript on page 3, lines 170-171 and in page 4, lines 222-227.
  • Comment 5.2: The mean age was 35.61 ± 12.50. It seems It seems there is significant variability in age. Were diagnoses of perimenopausal states taken into account?
  • Response 5.2: No, diagnoses of perimenopausal states were not specifically considered. Only patients with previous menstrual disorders were excluded, and we focused on women aged under 55 years. Age was used as an adjustment variable in the analyses to account for its variability.
  • Comment 5.3: The "incidence rate" is reported in section 3.1.6. Was the incidence rate actually calculated? From my point of view, there is confusion regarding the calculation of incidence in this study. What calculation was performed? Incidence rate? Cumulative incidence? This needs to be clarified. Additionally, neither of these should be presented as a percentage in the context of this study.
  • Response 5.3: Thank you for your observation. What we calculated was the incidence rate of new cases between doses. Due to the design of the study, cumulative incidences were not calculated. Instead, we focused on the incidence rate of new cases occurring between the doses of the vaccine. Even thought, as explained in the observation 4.3 of the methodology, we agree that the presentation of the incidence ratio was the best choice, so we recalculated to expressed by 100,000 habitants.
  • Comment 5.4: On the other hand, incidence calculations are performed, indicating a prospective temporality, but a logistic regression is conducted. Why not perform a Cox proportional hazards model and calculate the hazard ratio? These results would be more robust.
  • Response 5.4: Thank you for your insightful comment. We agree that a Cox proportional hazards model would provide robust results by accounting for time-to-event data and enabling the calculation of hazard ratios. However, our choice of logistic regression was based on the study design and data structure. Logistic regression was used to estimate the odds ratios because our primary data did not include precise timing of events, which is necessary for a Cox model. Additionally, methods like the self-controlled risk interval (SCRI) and self-controlled case series (SCCS) were considered for their ability to control for time-invariant confounders but could not be carried out due to the nature of our data. We acknowledge this limitation and have included a statement in the discussion.
  • Comment 5.5: It is also not clear in the results whether a risk measure (OR in this case) was calculated, adjusted for comorbidities and sociodemographic variables that could affect menstrual alteration. A table with the crude OR and adjusted OR values should be presented.
  • Response 5.5: Thank you for your comment. We agree with this observation. We have now included a table with crude OR values in the supplementary material. (Supplementary Table 3; Suplementary Table 4 – page 21)

Comment 6. Discussion: This section should be revised based on the clarifications of the previous doubts.

Response 6:  Thank you for your suggestions regarding the discussion section. I have carefully revised the discussion to incorporate the clarifications you provided. The limitations and potential biases are now more thoroughly addressed, ensuring that the study's conclusions are presented with the appropriate context and rigor. This change can be found in the revised manuscript on page 10, lines 663-679.

Reviewer 2 Report

Comments and Suggestions for Authors

Well-written manuscript and an interesting read! 
Just a few queies: 
Line 100: Women were excluded from the study if they had a diagnosis of any menstrual disorder before COVID-19 vaccination 101 and/or were pregnant and/or breastfeeding.

I wonder if PCOS should also be excluded as PCOS women usually would present with AM. Additionally, PCOS is often accompanied with certain degree of (pre-existing) long term health implication 

Secondly, how were HMB and AM defined in SIDIAP? 

Thank you 

Author Response

Comments:

Well-written manuscript and an interesting read! Just a few queies: 

Line 100: Women were excluded from the study if they had a diagnosis of any menstrual disorder before COVID-19 vaccination 101 and/or were pregnant and/or breastfeeding.
I wonder if PCOS should also be excluded as PCOS women usually would present with AM. Additionally, PCOS is often accompanied with certain degree of (pre-existing) long term health implication. Secondly, how were HMB and AM defined in SIDIAP? 

Thank you 

Response:

Thank you for this important observation. We agree that polycystic ovary syndrome (PCOS) is a significant condition that often overlaps with amenorrhea (AM) and has long-term health implications. However, instead of excluding women with PCOS, we included PCOS as a covariate in our association analyses to adjust for its potential confounding effects. This approach allows us to account for the presence of PCOS while examining the association between COVID-19 vaccination and menstrual disorders.

Additionally, the diagnostic codes for heavy menstrual bleeding (HMB) and amenorrhea (AM) in SIDIAP are based on the ICD-10-CM code list. Detailed information on these diagnostic codes can be found in Supplementary Table 1.

Reviewer 3 Report

Comments and Suggestions for Authors

Congratulations for this interesting important well written manuscript. 

But you are kindly requested to address the following comments: 

1. Further clarify how the Pfizer/BioNTech was associated with higher incidence of HMB and AM but not higher OR when exploring the effect of timing and doses. it is confusing and need to be explained in a better way 

2. Why Pfizer/BioNTech is not mentioned in tables 3 and 4? 

3. Why patient taking hormonal medications were not excluded?

Author Response

Congratulations for this interesting important well written manuscript. 

But you are kindly requested to address the following comments: 

Comments 1: Further clarify how the Pfizer/BioNTech was associated with higher incidence of HMB and AM but not higher OR when exploring the effect of timing and doses. it is confusing and need to be explained in a better way.

Response 1: Thank you for your observation. The Pfizer/BioNTech vaccine was associated with a higher incidence of HMB and AM due to the larger number of events observed in the vaccinated population. This is likely because it was the main vaccine administered in Europe. For this reason, we decided to use it as the comparator in the regression model.

Comments 2: Why Pfizer/BioNTech is not mentioned in tables 3 and 4? 

Response 2: The Pfizer/BioNTech vaccine was not mentioned in Tables 3 and 4 because it was used as the reference category. We have now included an explanation for this choice in the manuscript and added the Pfizer/BioNTech data for completeness and comparative purposes.

Comments 3: Why patient taking hormonal medications were not excluded?

Response 3: Thank you for your question. Patients taking hormonal medications were not excluded from the study because these medications can influence menstrual health and are therefore an important factor to consider in the analysis. Instead, the use of hormonal medications was included as a covariate in our regression models to adjust for their potential confounding effects. This approach allows us to better isolate the impact of COVID-19 vaccination on menstrual disorders.

Round 2

Reviewer 1 Report

Comments and Suggestions for Authors

Many thanks to the authors for having given such a deep, detailed and well-worked response to my previous doubts, especially regarding the methodological doubts. The authors have improved the scientific quality of this work based on their answers. Thank you very much.